# Assessing the Predictive Accuracy of EORTC, CUETO and EAU Risk Stratification Models for High-Grade Recurrence and Progression after Bacillus Calmette–Guérin Therapy in Non-Muscle-Invasive Bladder Cancer

**DOI:** 10.3390/cancers16091684

**Published:** 2024-04-26

**Authors:** Aleksander Ślusarczyk, Karolina Garbas, Patryk Pustuła, Łukasz Zapała, Piotr Radziszewski

**Affiliations:** Department of General, Oncological and Functional Urology, Medical University of Warsaw, 02-005 Warsaw, Poland

**Keywords:** non-muscle-invasive bladder cancer, progression, high-grade recurrence, BCG, EORTC, risk model

## Abstract

**Simple Summary:**

We aimed to identify risk factors and evaluate the accuracy of existing risk stratifications developed for non-muscle-invasive bladder cancer (NMIBC) regarding their ability to predict high-grade recurrence and progression. We included 171 NMIBC patients treated with TURBT and adjuvant BCG, of whom 73 experienced recurrence (42.7%), and 29 developed progression (17%). Available risk models (EORTC/ CUETO/ EAU) demonstrated limited accuracy in predicting high-grade recurrence-free (RFS) and progression-free survival (PFS). Multivariable analysis identified independent predictors for high-grade RFS, including T1HG tumor at repeat TURBT, tumor multiplicity, previous history of high-grade NMIBC, and EORTC2006 progression risk score. In conclusion, the available risk models lack accuracy in predicting high-grade RFS and PFS in BCG-treated NMIBC, suggesting potential improvement with the inclusion of additional risk factors.

**Abstract:**

The currently available EORTC, CUETO and EAU2021 risk stratifications were originally developed to predict recurrence and progression in non-muscle-invasive bladder cancer (NMIBC). However, they have not been validated to differentiate between high-grade (HG) and low-grade (LG) recurrence-free survival (RFS), which are distinct events with specific implications. We aimed to evaluate the accuracy of available risk models and identify additional risk factors for HG RFS and PFS among NMIBC patients treated with Bacillus Calmette–Guérin (BCG). We retrospectively included 171 patients who underwent transurethral resection of the bladder tumor (TURBT), of whom 73 patients (42.7%) experienced recurrence and 29 (17%) developed progression. Initially, there were 21 low-grade and 52 high-grade recurrences. EORTC2006, EORTC2016 and CUETO recurrence scoring systems lacked accuracy in the prediction of HG RFS (C-index 0.63/0.55/0.59, respectively). EAU2021 risk stratification, EORTC2006, EORTC2016, and CUETO progression scoring systems demonstrated low to moderate accuracy (C-index 0.59/0.68/0.65/0.65) in the prediction of PFS. In the multivariable analysis, T1HG at repeat TURBT (HR = 3.17 *p* < 0.01), tumor multiplicity (HR = 2.07 *p* < 0.05), previous history of HG NMIBC (HR = 2.37 *p* = 0.06) and EORTC2006 progression risk score (HR = 1.1 *p* < 0.01) were independent predictors for HG RFS. To conclude, available risk models lack accuracy in predicting HG RFS and PFS in -NMIBC patients treated with BCG.

## 1. Introduction

Bladder cancer is a heterogeneous disease and three-quarters of patients are diagnosed at an early stage [1,2]. Non-muscle-invasive bladder cancer (NMIBC) denotes tumors confined to mucosa and submucosa, which might be effectively treated with transurethral resection of the bladder tumor (TURBT) although recurrences are common [1,2]. Intravesical instillations with Bacillus Calmette–Guérin (BCG) are recommended to reduce the recurrence risk in intermediate- and high-risk NMIBC [1]. Recurrences can lead to progression and worsen further prognoses, and, even if benign, can substantially affect the quality of life. Our recent population-based study showed that cancer-specific deaths are not uncommon in long-term follow-up of high-grade NMIBC, reaching up to 19% of high-grade T1 tumors [3]. Multiple prognostic stratification tools were established to facilitate the choice of optimal risk-adapted adjuvant therapy and to tailor the follow-up [4,5,6,7,8]. Nomograms and risk-scoring models were developed in different cohorts of patients and external validations reveal their limitations in discriminative power and calibration [9,10,11,12].

European Organization for Research and Treatment of Cancer (EORTC) 2006 risk tables were established in the general cohort of NMIBC without adjuvant BCG. Spanish Urological Club for Oncological Treatment (CUETO) risk tables were developed in the cohort of patients who received short-term BCG therapy. The EORTC 2016 nomogram was created for the intermediate and high-risk NMIBC patients treated with maintenance BCG [6,7,8]. Finally, the European Association of Urology (EAU) 2021 risk stratification was founded to determine progression risk in the group of primary NMIBC, who did not receive BCG [5]. Importantly, none of the abovementioned risk models incorporated the information from repeat transurethral resection (reTUR) and reTUR was not routinely performed in studies resulting in the development of the EORTC 2006, CUETO and EORTC 2016 [6,7,8]. A recent meta-analysis reported the contemporary prognostic role of reTUR and its association with recurrence-free survival [13]. External validation of EORTC 2006 and CUETO revealed its unsatisfactory accuracy and discrimination in the retrospective analysis of 4689 patients with NMIBC [9]. Another external validation of available nomograms by Krajewski et al. demonstrated an overestimation of progression risk and low discrimination for recurrence, when using CUETO and EAU 2021 risk models in NMIBC treated with routine reTUR and adequate BCG [10]. Validation in the cohort of high-risk BCG-exposed patients revealed an overestimation of progression risk for the updated EAU 2021 model [11].

Notably, none of the aforementioned nomograms was designed or validated to assess the risk of high-grade (HG) recurrences, which have distinct prognoses and implications compared to low-grade (LG) recurrences [14,15]. Approximately 30% of recurrences occurring in patients treated with BCG are of low grade, which is not considered therapy failure and should not prompt the cessation of the treatment [14,15]. Only high-grade recurrence during BCG (or progression) meets BCG-unresponsive criteria and warrants discontinuation of further BCG instillations [1,15,16].

In this study, we aimed to identify the risk factors for high-grade recurrence and progression among BCG-treated patients with intermediate-, high- and very-high-risk NMIBC and to validate the accuracy of currently used risk stratifications.

## 2. Materials and Methods

### 2.1. Study Design and Selection Criteria

This is a retrospective, single-tertiary-center study. Inclusion criteria included the presence of intermediate-, high-, and very-high-risk NMIBC in adult patients, treated with TURBT between 2010 and 2019 who received subsequent intravesical BCG instillations. Exclusion criteria encompassed lack of adequate induction course of BCG defined as at least 5 of 6 instillations (*n* = 11), delayed BCG therapy > 4 months after the last TURBT (*n* = 7), presence of isolated carcinoma in situ (CIS/Tis) without concomitant papillary tumor (*n* = 27).

### 2.2. Treatment and Follow-up

All patients underwent TURBT at our department. TURBT was performed with the patient in a lithotomy position under spinal or general anesthesia in accordance with EAU clinical guidelines [1]. A resection loop with monopolar current was utilized. All patients who received incomplete initial TURBT underwent repeat TURBT (also called as second or restaging transurethral resection). ReTUR was performed whenever indicated by EAU clinical guidelines or upon the treating physician’s decision [1]. 

Surgical specimens were reviewed by a genitourinary pathologist, graded according to the 1973 and 2004 WHO grading systems and staged according to the 2009 TNM classification. 

Patients with high-grade or T1-stage tumors or CIS or multiple and recurrent low-grade Ta tumors were qualified for BCG in the standard schedule. All patients received an induction BCG course of at least 5 of 6 weekly intravesical instillations [1]. The maintenance schedule included 3 weekly instillations at 3, 6, 12, 18, 24, 30 and 36 months. Adequate BCG was defined as at least 5 of 6 instillations of induction and 2 of 3 instillations of the first maintenance course [1,17]. Full-dose of BCG was administered including at least 2 × 10^8^ and no more than 3 × 10^9^ viable units. RIVM strain was used in 98.2% of patients.

Follow-up involved regular cystoscopy and urine cytology performed every 3 months in the first two years and every six months from the second to the fifth year [1,17]. Any suspicion of recurrence or progression was verified every time with TURBT. 

### 2.3. Outcomes

High-grade recurrence-free survival was the primary outcome. Low-grade recurrence-free survival, recurrence-free survival, and progression-free survival were secondary outcomes. Progression was defined as the occurrence of muscle-invasive bladder cancer or the development of distant metastasis. High-grade recurrence was defined as any grade 3 or high-grade recurrence following BCG. Low-grade recurrence was defined as initial grade 1/2 or low-grade papillary tumor recurrence following BCG. Salvage radical cystectomy was performed in eligible patients upon progression or high-grade recurrence. 

Survival time was calculated from the date of index TURBT to the event of interest. Patients were censored at the date of last follow-up, death due to any cause or the date of salvage radical cystectomy. To estimate PFS following LG and HG recurrences, the date of recurrence was used as the index point in survival analysis. 

### 2.4. Ethics Statement

Due to the character of this study, the Institutional Review Board waived the need for study approval. This study was performed in accordance with the Declaration of Helsinki and its later amendments.

### 2.5. Statistical Analysis

Descriptive variables included clinical, histopathological and survival data. Histopathological data included primary staging, grading, presence of concomitant CIS and the pathology at reTUR. Clinical data included previous patterns of recurrence (HG/LG/none and its frequency per year), tumor size, multifocality, age, gender and comorbidities summarized within Charlson comorbidity index as previously [18]. 

Patients were risk stratified using EAU 2021, EAU 2019, AUA, EORTC 2006, EORTC 2016 and CUETO risk models and scores [5,6,7,8,19]. Dedicated recurrence and progression risk scores were calculated if applicable. Risk stratification were performed based on the clinical and histopathological data available at index TURBT.

Validation of risk models was performed with Cox proportional hazards. Discrimination was assessed using the concordance index (C-index) and area under the ROC curve (AUC). For calibration, we compared the actual estimates from Kaplan–Meier curves and expected survival rates predicted by the respective risk models.

The Kaplan–Meier method was used to compute survival estimates within 1- and 5-year time points. Reverse Kaplan–Meier methods were used to estimate the median follow-up with interquartile ranges (IQR). Cox proportional hazard (CPH) regressions were used for survival analysis. Univariable and multivariable Cox proportional hazards analyses were performed. Multivariable analyses included only selected variables based on the univariable analyses. Stepwise selection of variables was applied. Hazard ratios along with 95% confidence intervals (95% CI) were derived from CPH regression. 

In order to internally validate the performance of our risk model and mitigate the potential effects of overfitting, we employed a bootstrap resampling technique. Specifically, we generated 300 bootstrap samples, each drawn with replacement from the original dataset while maintaining the same sample size. Optimism for the C-index was calculated.

Continuous variables were presented as median values accompanied by ranges between quartiles. Differences between groups were evaluated with the U-Mann–Whitney test for continuous variables and with Fischer’s exact test or Chi-square test for categorical variables. For all statistical analyses, a two-sided *p*-value < 0.05 was considered statistically significant. SAS software version 9.4 (SAS Institute, Cary, NC, USA) was used to conduct statistical analysis.

## 3. Results

Two hundred and sixteen consecutive patients with intermediate-, high-, and very-high-risk NMIBC, who underwent TURBT between 2010 and 2019 and subsequently received BCG instillations, were identified in our institutional database. We included 171 patients who met the inclusion criteria. Of these, 128 were males (75%) and 43 were females (25%), with a median age of 71 years (IQR 64–79). The cohort predominantly consisted of patients with T1HG (*n* = 118; 69%), followed by TaHG (*n* = 26; 15.2%), T1LG (*n* = 9; 5.3%), and TaLG (*n* = 18; 10.5%) tumors. The median survival follow-up was 65 (IQR 24–90) months. Detailed baseline characteristics are presented in Table 1.

There were 31 patients with IR (18.1%), 106 with HR (62%) and 34 with VHR (19.9%) NMIBC, according to EAU 2021 risk stratification. ReTUR was performed in 126 patients (73.7%), of whom 60 had residual cancer, including 35 papillary tumors (20.5%) and 25 CIS (14.6%). De novo detection of CIS in reTUR was found in 8 patients (6.3% among those treated with reTUR). Adequate BCG was administered to 132 patients (77.2%) and 74 individuals (43.3%) received at least a 1-year maintenance BCG schedule. 

### 3.1. Oncological Outcomes

Overall, 73 patients (42.7%) experienced recurrence and 29 (17%) developed progression to MIBC following BCG therapy. Comparison of baseline characteristics between patients with initial LG and HG recurrence are presented in the Appendix A Table A1. In our cohort, 5-year estimates of PFS, HG RFS and RFS were 81.4%, 65.2% and 53.7%, respectively. Overall, 21 recurrences were initially low-grade and 52 were initially high grade. Kaplan–Meier curves illustrating PFS, RFS, HG RFS and LG RFS are presented in Figure 1.

### 3.2. Significance of Low-Grade and High-Grade Recurrences

After initial low-grade recurrence, four patients (19.1%) developed high-grade recurrence and three (14.3%) progressed to MIBC. Among patients with initial high-grade recurrence, 26 (50%) developed MIBC, and 9 (17.3%) underwent salvage cystectomy for recurrent NMIBC. The median time from initial high-grade recurrence to subsequent MIBC progression was 9.6 months. Estimates of 5-year PFS after HG recurrence were 33.9% and after LG recurrence were 88.3%.

### 3.3. Novel EAU 2021 Risk Stratification

The EAU2021 risk stratification successfully grouped patients according to progression risk, which were 3.5%, 20%, and 25.8% in IR, HR and VHR groups at 5-year follow-up. Comparison of these estimates to reference risks reported in the EAU 2021 risk tables indicates poor calibration of the model, with underestimation of risk in HR (estimated risk of 9.6–11%) and overestimation in VHR groups (estimated risk of 40–44%). 

EAU2021 risk stratification successfully stratified patients according to high-grade recurrence risk, which was 13.8%, 35.2%, and 48.5% in IR, HR and VHR groups at 5 years. As EAU 2021 did not report the HG recurrence risks we were unable to validate its calibration in that setting.

### 3.4. Validation of Available Risk Models

EAU 2021 risk stratification, EORTC 2006, EORTC 2016 and CUETO recurrence scoring systems lacked accuracy in the prediction of high-grade recurrence (C-index 0.57/0.63/0.55/0.59, respectively). EAU 2021 risk stratification, EORTC 2006, EORTC 2016, CUETO progression scoring systems demonstrated low to moderate accuracy (C-index 0.59/0.68/0.65/0.65) in predicting progression to MIBC. Detailed analyses of the accuracy of available risk models for the prediction of RFS, LG RFS, HG RFS and PFS are presented in Table 2.

### 3.5. Multivariable Analyses for RFS, HG RFS and PFS

Univariable analyses were performed with Cox proportional hazards to predict HG RFS and PFS and served for the optimal choice of risk factors for multivariable analyses (Table 3).

In the multivariable analysis with Cox proportional hazards, EORTC 2006 progression risk score (HR = 1.1 95% CI 1.03–1.19 *p* < 0.01), T1HG at reTUR (HR = 3.17 95% CI 1.48–6.79 *p* < 0.01), tumor multiplicity (HR = 2.07 95% CI 1.10–3.93 *p* < 0.05) and previous history of HG NMIBC (HR = 2.37 95% CI 0.96–5.86 *p* = 0.06) were independent predictors for high-grade RFS (Table 4 A). Our model was characterized by a C-index of 0.74 with a 1- and 5-year AUC of 0.78 and 0.78, respectively. Internal validation of the model with 300 bootstrapped samples revealed slight overfitting with an optimism correction of 0.04.

In the multivariable analysis with Cox proportional hazards, EORTC 2006 recurrence risk score (HR = 1.19 95% CI 1.09–1.29 *p* < 0.001) and T1HG at reTUR (HR = 2.67 95% CI 1.39–5.15 *p* < 0.01) constituted independent risk factors for any RFS (Table 4 B).

In the multivariable analysis with Cox proportional hazards, T1G3 tumor (HR = 5.7 95% CI 1.20–27.2 *p* < 0.05), multiplicity (HR = 4.1 95% CI 1.67–10.0 *p* < 0.01), presence of HG T1 at reTUR (HR = 3.86 95% CI 1.48–10 *p* < 0.01) and previous history of HG tumor (HR = 10.3 1.76–60 *p* < 0.05) were independent predictors for PFS (Table 4 C). Our model was characterized by a C-index of 0.79 with a 1- and 5-year AUC of 0.77 and 0.85, respectively. Internal validation of the model with 300 bootstrapped samples revealed slight overfitting with an optimism correction of 0.046.

## 4. Discussion

In this study, we validated the currently used risk classifications and proposed our risk model for the prediction of high-grade recurrence in BCG-treated patients with NMIBC.

We found that existing risk stratifications and models for NMIBC recurrence and progression lack accuracy in the population of patients with predominantly high- and very-high-risk NMIBC who receive BCG. Additionally, initial high-grade recurrences were more frequent and conferred a very high progression risk (5-year PFS of 34%), whereas initial low-grade recurrence, although not benign, preceded progressions only in the minority of such cases (5-year PFS 88%). There is a need for dedicated risk models specifically tailored to assess the risk of high-grade recurrence, as high- and low-grade recurrences differ in terms of further prognosis and implications for treatment.

Moreover, none of the available stratifications was constructed or validated to predict high-grade recurrence but rather designed to predict any recurrence. Our study showed the poor accuracy of EORTC 2006, EORTC 2016 and CUETO for the prediction of high-grade recurrence (C-indices 0.55–0.63). We found that the novel EAU 2021 risk model could be used not only for the stratification of progression risk but also for high-grade recurrence risk, albeit with poor accuracy for both events (C-indices of 0.57 and 0.59, respectively). We identified several risk factors that have not been used in any of the available stratification tools but appear to be significant predictors for high-grade recurrence. Our risk model included reTUR pathology and previous grade pattern that constituted adjunct to EORTC progression risk score and tumor multiplicity. 

Furthermore, none of the available risk-scoring models provided higher accuracy for progression risk assessment than the set of four risk factors: the presence of T1G3, tumor multiplicity, presence of residual T1HG at reTUR and previous history of HG tumor. However, due to the relatively small sample size, validation in larger datasets is required. In our analysis, PFS was overestimated in the EAU 2021 HR group and underestimated in the VHR cohort.

We believe that dedicated risk models for the prediction of high-grade recurrence in BCG-treated populations can be clinically useful when counseling patients for further management after TURBT for high-grade NMIBC. To date, such models are not available and widely recognized EORTC 2016 and CUETO recurrence risk tables are recommended by the EAU guidelines for the estimation of risk for any recurrence in BCG-treated NMIBC [1]. However, as we have demonstrated, the discrimination and accuracy of such models are relatively low. Interestingly, CUETO and EORTC progression risk scores exhibit higher accuracy than dedicated CUETO or EORTC recurrence risk scores in predicting high-grade recurrence. This can be explained by the difference in weights of cardinal risk factors such as tumor category, grade, and presence of CIS, which contribute to higher CUETO/ EORTC progression risk scores, but do not highly affect CUETO/ EORTC recurrence risk scores [6,8]. In our multivariable analysis, the EORTC progression risk score was selected as the most significant among other risk scores and entered the model for HG recurrence risk. Other risk factors include pathology at reTUR with the presence of residual HG T1 as a strong adverse feature. High-grade T1 at reTUR was previously reported as associated with very unfavorable 5-year RFS (18%) and PFS (52%) among BCG-treated patients [20]. 

We are concerned about not including reTUR for prognostic purposes in any of the available risk models. In the study that developed a novel EAU 2021 risk model (22% of patients with T1) reTUR was performed in 16% of patients, whereas in EORTC 2006, EORTC 2016 and CUETO population reTUR was not routinely performed [6,7,8]. Importantly reTUR could result in a change in the risk score (e.g., detection of CIS) and ensure completeness of prior resection. In our study, reTUR was performed in 73.8% of patients, residual papillary tumors at reTUR were found in 27.7% of patients in whom reTUR was performed and primary detection of CIS previously not biopsied during the index TURBT was found in 6.3% of patients in whom reTUR was performed.

Previous history of high-grade tumors compared to previous low-grade was independently associated with an increased risk for further high-grade recurrence. A study by Thomas has previously shown that among high-risk patients, a previous history of high-grade tumors was associated with a higher risk of progression to MIBC, compared to a history of progressive low-grade and primary tumors [21]. Despite its inclusion in the calculation of the EORTC score, multiplicity appeared as an independent risk factor in the multivariable analysis. 

Our risk model was characterized by an acceptable accuracy, with a C-index of 0.74, and did not reveal the significant risk of overfitting. Such a model can serve as guidance in the patient’s counseling before BCG therapy. Our model mostly identifies patients who will be considered as BCG-unresponsive. Therefore, perhaps patients with T1HG at reTUR, with previous high-grade tumor, with multiple lesions and higher EORTC progression risk scores should be offered an enrolment in clinical trials aiming at improving response to BCG [2]. The majority of high-grade recurrences result in BCG unresponsiveness except for late relapses after BCG interruption (>6 months) and papillary Ta or CIS before maintenance BCG administration [4,16]. In our previous paper, we showed that inflammatory markers could be used as predictors of BCG-unresponsive disease [22].

The primary challenge and ultimate goal of further updating the EAU 2021 risk stratification was to identify patients who will progress [5]. Such a group is at the highest risk for cancer-specific death which could be prevented by immediate or early cystectomy. The novel EAU 2021 risk model successfully identified a group of patients with a 40% risk of progression at 5-year follow-up [5]. However, external validations of the EAU 2021 risk model underscored the overestimation of risk in BCG-treated high- and very-high-risk patients [10,11]. In our cohort, progression risk overestimation within the use of EAU 2021 was also observed but only for the VHR group and not for the HR group in which the risk was actually underestimated. We identified the presence of T1G3, tumor multiplicity, presence of residual T1HG at reTUR and previous history of HG tumor as independent risk factors for progression. It is already clear that high-grade T1 tumors are most likely to progress among other NMIBC and are associated with undeniable long-term cancer-specific mortality [18]. This is raised as an argument for early radical cystectomy to prevent the progression and its fatal consequences [23]. 

Low-grade recurrences during BCG therapy were reported in a few other papers. A study by Li et al. demonstrated that the grade of tumor recurrence following intravesical BCG treatment serves as a crucial indicator for predicting the progression of bladder cancer to muscle-invasive or metastatic urothelial carcinoma [14]. Although individuals experiencing low-grade recurrences have fewer progression events, compared to those with high-grade recurrences, their estimated 5-year progression rate was still 14.4% in that study [14]. Our study confirmed that low-grade recurrence can precede high-grade recurrence and progression which in our population occurred relatively late in the follow-up. Nevertheless, progression risks are significantly lower for LG than for HG recurrence and the presence of LG recurrence does not meet the BCG unresponsive criteria and is not an indication for BCG interruption [4,15].

We anticipate the imminent update of current risk models as our understanding of the role of the urinary microbiome expands, and as urine- and blood-based biomarkers are developed [2]. Recent studies have underscored the potential significance of the urinary microbiome in the detection and course of NMIBC, sparking further interest in the investigation [24,25]. However, novel blood-based and easily accessible biomarkers, such as systemic immune-inflammatory markers and the well-recognized neutrophil-to-lymphocyte ratio, were not validated in this study due to their absence in the current clinically utilized risk models [22,26]. The suboptimal accuracy of existing models and the lack of inclusion of newly developed and potentially significant prognostic factors highlight the limitations of these models and emphasize the imperative for the development of new, more comprehensive risk assessment tools.

Limitations of our study come from the inherent nature of its retrospective single-center character and low sample size. Our cohort did not include patients with isolated CIS without concomitant papillary tumors. We decided to exclude these patients as they were also excluded from studies developing EORTC 2016 and EAU 2021 risk models [5,7]. Information regarding smoking was not available for all patients and was therefore not included in the regression analysis despite the recent evidence for the impact of smoking on RFS and PFS [27]. Another limitation that must be acknowledged is the suboptimal regimen duration, which nonetheless reflects real-world clinical practice. Eleven patients received only 5 out of 6 instillations of induction BCG due to adverse events. In 6 of these 11 patients, BCG was continued in further maintenance instillations as adverse events resolved. Another important issue, considering the recommended BCG regimen duration, is the low percentage (13.5%) of patients who received the 3-year maintenance schedule. Notably, even in RCTs like the SWOG study, which showed the superiority of maintenance BCG over induction alone, only 16% of patients completed the 3-year maintenance schedule [17]. On the other hand, in the EORTC-GU Cancers Group Randomized Study, 35% of patients completed the 3-year full BCG maintenance regimen to which they were allocated [28]. Our results reflect real-world treatment patterns, which provide valuable validation of available risk models, and such validation is necessary to ensure the applicability of the risk models beyond a clinical trial setting.

Furthermore, future studies could incorporate emerging urine biomarkers and extended pathological assessment of TUR specimens including T1 sub-staging and immune-related gene expression to refine predictive models and enhance their clinical utility [2,29]. Additionally, conducting multicenter validation studies will be imperative to confirm the generalizability and reliability of our findings across diverse clinical settings and within larger cohorts.

## 5. Conclusions

To conclude, available risk models lack accuracy in predicting high-grade RFS and PFS in NMIBC patients treated with BCG. High- and low-grade recurrences have distinct prognosis and treatment implications. We found that among different risk models, the EORTC progression score had the highest accuracy for the prediction of high-grade recurrence. Pathology at reTUR, previous history of high-grade NMIBC and tumor multiplicity provided additional prognostic information. Further studies are required to improve existing risk models for high-risk NMIBC treated with BCG.

## Figures and Tables

**Figure 1 cancers-16-01684-f001:**
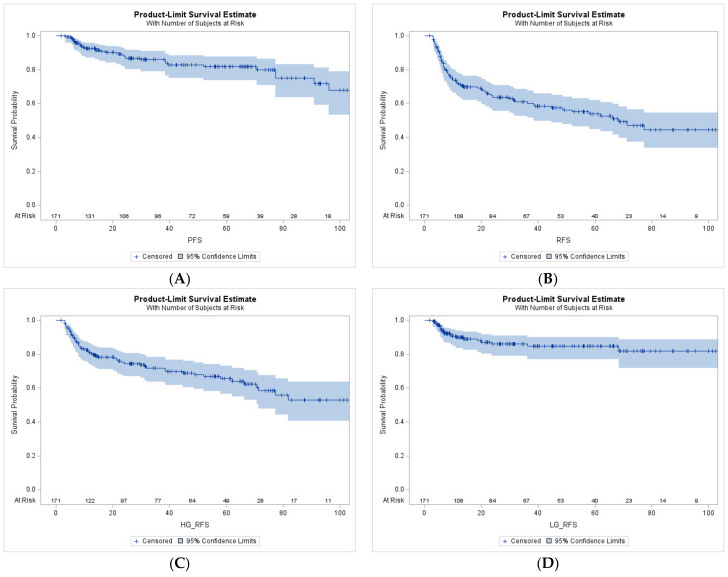
Kaplan–Meier curves representing PFS (**A**), RFS (**B**), HG RFS (**C**), and LG RFS (**D**).

**Table 1 cancers-16-01684-t001:** Baseline characteristics of included patients with NMIBC treated with BCG.

Variables		Whole Cohort	
		Number of Patients/Median	% of Patients/IQR
Stage	Ta	44	25.73
	T1	127	74.27
Tumor Grade WHO 2004	HG	147	85.96
	LG	24	14.04
Tumor Grade WHO 1973	G1	7	4.09
	G2	28	16.37
	G3	136	79.53
Concomitant CIS	No	149	87.13
	Yes	22	12.87
Multiplicity	No	79	46.20
	Yes	92	53.80
Tumor size	≤3 cm	109	63.74
	>3 cm	62	36.26
Previous recurrence frequency	None	114	66.67
	<1/year	39	22.81
	≥1/year	18	10.53
Previous history of NMIBC	High-grade tumor	18	10.53
	Low-grade tumor	39	22.81
	Primary	114	66.67
Detrusor muscle in index TURBT	Yes	127	74.27
	No	44	25.73
ReTURBT pathology	T0	66	38.60
	T1HG	26	15.20
	TaHG	4	2.34
	TaLG	5	2.92
	Tis	25	14.62
	none	45	26.32
EAU 2021 progression risk groups	IR	31	18.13
	HR	106	61.99
	VHR	34	19.88
	IR	12	7.02
EAU 2019 risk groups	HR	72	42.11
	VHR	87	50.88
EORTC 2016 progression risk	T1G3	105	61.40
	T1G2/TaG3	52	30.41
	T1G1/TaG2	8	4.68
	TaG1	6	3.51
EORTC 2016 recurrence risk	Group 1	126	73.68
	Group 2	27	15.79
	Group 3	7	4.09
	Group 4	11	6.40
AUA risk groups	IR	28	16.37
	HR	143	83.63
Age	Years	71	64–79
Gender	Female	43	25.15
	Male	128	74.85
Charlson comorbidity score	Median	6	4–7
EORTC 2006 recurrence risk score	Median	6	5–8
EORTC 2006 progression risk score	Median	12	8–14
CUETO recurrence risk score	Median	6	5–9
CUETO progression risk score	Median	9	8–10

IQR—interquartile range; HG—high grade; LG—low grade; CIS—carcinoma in situ; TURBT—transurethral resection of bladder tumor; ReTURBT—repeat (restaging) transurethral resection of bladder tumor; IR—intermediate risk; HR—high risk; VHR—very-high risk; EAU—European Association of Urology; AUA—American Urological Association; EORTC2016 groups 1–4 classified according to recurrence rate and multiplicity [7].

**Table 2 cancers-16-01684-t002:** Discrimination and accuracy of available risk models for recurrence and progression in NMIBC.

Risks Model		EORTC 2006 Recurrence Risk Score	EORTC 2006 Progression Risk Score	CUETO Recurrence Risk Score	CUETO Progression Risk Score	EAU 2021	EAU 2019	EORTC 2016 Recurrence Grouping	EORTC 2016 Progression Grouping	AUA
Any recurrence	C-index	0.63	0.61	0.58	0.60	0.52	0.55	0.57	0.53	0.51
	AUC 1 year	0.63	0.62	0.60	0.59	0.53	0.54	0.59	0.52	0.53
	AUC 5 years	0.65	0.66	0.55	0.64	0.55	0.58	0.58	0.55	0.54
	*p*-value	<0.001	<0.001	0.01	0.002	0.54	0.19	0.076	0.39	0.36
High-grade recurrence	C-index	0.63	0.66	0.59	0.65	0.57	0.60	0.55	0.58	0.56
	AUC 1 year	0.64	0.66	0.63	0.60	0.55	0.58	0.56	0.56	0.56
	AUC 5 years	0.67	0.74	0.60	0.70	0.61	0.64	0.55	0.60	0.60
	*p*-value	<0.001	<0.001	0.015	<0.001	0.063	0.033	0.22	0.16	0.045
Low-grade recurrence	C-index	0.60	0.50	0.57	0.52	0.60	0.60	0.62	0.60	0.55
	AUC 1 year	0.54	0.49	0.54	0.58	0.56	0.60	0.67	0.56	0.57
	AUC 5 years	0.53	0.57	0.46	0.50	0.60	0.62	0.57	0.58	0.58
	*p*-value	0.089	0.81	0.40	0.63	0.26	0.17	0.16	0.19	0.10
Progression	C-index	0.62	0.68	0.53	0.65	0.59	0.65	0.49	0.66	0.58
	AUC 1 year	0.66	0.68	0.54	0.60	0.56	0.62	0.45	0.64	0.59
	AUC 5 years	0.64	0.70	0.48	0.67	0.61	0.65	0.46	0.66	0.60
	*p*-value	0.007	0.015	0.47	0.039	0.26	0.028	0.95	0.21	0.13

AUC 1 year—Area Under the Curve regarding 1-year event-free survival; AUC 5 year–Area Under the Curve regarding 5-year event-free survival; *p*-value is calculated for univariable Cox proportional hazard for selected risk model.

**Table 3 cancers-16-01684-t003:** Univariable analyses with Cox proportional hazards for predicting high-grade recurrence-free survival and progression-free survival.

Variable		Univariable Analysis for HG RFS			Univariable Analysis for PFS		
		HR	95% CI	*p*-value	HR	95% CI	*p*-value
Stage	Ta	ref			ref		
	T1	1.560	0.78–3.09	0.20	2.753	0.83–9.13	0.10
Tumor grade WHO 2004	LG	ref	-		ref	-	
	HG	2.988	0.93–9.57	0.065	4.356	0.59–32.1	0.15
Concomitant CIS	No		-		ref	-	
	Yes	1.769	0.91–3.42	0.09	0.529	0.12–2.22	0.39
Multiplicity	No	ref	-		ref	-	
	Yes	2.963	1.62–5.39	<0.001	3.256	1.38–7.65	0.007
Tumor size	≤3cm	ref	-		ref	-	
	>3cm	0.865	0.49–1.51	0.61	1.434	0.68–2.98	0.33
Previous history of NMIBC	LG	ref	-		ref	-	
	HG	2.580	1.09–6.10	0.031	6.443	1.29–32.0	0.023
	Primary	1.010	0.51–1.99	0.97	3.468	0.81–14.8	0.093
Recurrent tumor	No	ref	-		ref	-	
	Yes	1.394	0.80–2.40	0.23	0.790	0.34–1.78	0.57
Detrusor Muscle in the index TURBT	Yes	ref	-		ref	-	
	No	1.124	0.62–2.02	0.69	0.749	0.30–1.84	0.53
ReTURBT pathology	T0	ref	-		ref	-	
	T1HG	2.922	1.37–6.19	0.005	3.240	1.27–8.25	0.014
	TaHG	-	-	0.98	-	-	0.99
	TaLG	1.086	0.14–8.21	0.93	-	-	0.99
	Tis	2.336	1.05–5.18	0.036	1.636	0.50–5.29	0.41
	None	2.111	1.05–4.23	0.035	1.365	0.51–3.59	0.53
Age	Years	1.019	0.99–1.04	0.19	1.028	0.98–1.06	0.17
Charlson comorbidity score	Points	1.207	1.04–1.39	0.009	1.222	1.00–1.48	0.046
Gender	Female	ref	-		ref	-	
	Male	1.077	0.58–1.97	0.81	3.287	0.99–10.8	0.051
EORTC 2006 recurrence score	points	1.191	1.08–1.30	<0.001	1.193	1.05–1.35	0.007
EORTC 2006 progression score	points	1.151	1.07–1.23	<0.001	1.123	1.02–1.23	0.015
CUETO recurrence risk score	points	1.120	1.02–1.22	0.012	0.954	0.83–1.09	0.49
CUETO progression risk score	points	1.243	1.10–1.40	<0.001	1.190	1.00–1.40	0.039
EAU 2021 progression risk groups	IR	ref	-		ref	-	
	HR	1.872	0.73–4.78	0.19	2.410	0.55–10.3	0.24
	VHR	3.094	1.13–8.45	0.028	3.529	0.74–16.6	0.11
EAU 2019 risk groups	IR	ref	-		ref	-	
	HR	0.310	0.04–2.32	0.25	1.057	0.12–8.80	0.95
	VHR	1.831	1.03–3.22	0.036	3.217	1.30–7.94	0.011
EORTC 2016 for recurrence	Group 1	ref	-		ref	-	
	Group 2	1.784	0.87–3.62	0.11	1.111	0.38–3.24	0.037
	Group 3	1.086	0.26–4.52	0.91	1.095	0.14–8.22	0.008
	Group 4	2.017	0.85–4.78	0.11	1.195	0.28–5.10	0.057
EORTC 2016 for progression	TaG3/T1G2	ref	-		ref	-	
	T1G3	1.794	0.94–3.41	0.07	3.042	1.05–8.79	0.039
	T1G1/TaG2	0.464	0.06–3.57	0.46	1.542	0.17–13.8	0.69
	TaG1	0.805	0.11–6.2	0.84	-	-	-
AUA risk stratification	IR	ref	-		ref	-	
	HR	3.290	1.02–10.5	0.045	4.716	0.63–34.8	0.13

HG—high grade; LG—low grade; Tis—tumor in situ (CIS), TURBT—transurethral resection of bladder tumor; ReTURBT—repeat (restaging) transurethral resection of bladder tumor; IR—intermediate risk; HR—high risk; VHR—very-high risk; EAU—European Association of Urology; AUA—American Urological Association; HR—hazard ratio; CI—confidence interval.

**Table 4 cancers-16-01684-t004:** Multivariable analyses with Cox proportional hazards for predicting high-grade recurrence-free survival (A), recurrence-free survival (B) and progression-free survival (C).

A. High-grade recurrence-free survival.					
Variable		HR	95% CI		*p*-value
EORTC 2006 progression risk score	points	1.103	1.027	1.185	0.007
	T0	ref			
	T1HG	3.174	1.484	6.786	0.003
	TaHG	-	-	-	-
ReTURBT pathology	TaLG	0.989	0.130	7.509	0.99
	Tis	2.217	0.989	4.971	0.053
	not performed	2.034	0.910	4.550	0.084
Multiplicity	no	ref			
	yes	2.071	1.093	3.925	0.026
Previous history of NMIBC	low-grade tumor	ref			
	high-grade tumor	2.370	0.959	5.857	0.061
	primary tumor	1.361	0.612	3.024	0.45
B. Recurrence-free survival.					
Variable		HR	95% CI		*p*-value
EORTC 2006 recurrence risk score	points	1.185	1.085	1.293	<0.001
ReTURBT pathology	T0	ref			
	T1HG	2.672	1.387	5.147	0.003
	TaHG	0.728	0.098	5.408	0.75
	TaLG	0.480	0.064	3.589	0.47
	Tis	2.130	1.040	4.363	0.039
	not performed	1.916	0.947	3.876	0.07
Stage	Ta	ref			
	T1	1.696	0.860	3.342	0.12
C. Progression-free survival.					
Variable		HR	95% CI		*p*-value
Stage and grade	TaG3	ref			
	T1G3	5.703	1.197	27.176	0.028
	T1G1/G2	1.252	0.167	9.372	0.83
	TaG1/G2	2.838	0.206	39.076	0.43
ReTURBT pathology	T0	ref			
	T1HG	3.856	1.483	10.024	0.006
	TaHG	-	-	-	-
	TaLG	-	-	-	-
	Tis	2.149	0.637	7.258	0.22
	not performed	3.194	0.987	10.334	0.053
Multiplicity	no	ref			
	yes	4.092	1.670	10.025	0.002
Previous history of NMIBC	low-grade tumor	ref			
	high-grade tumor	10.292	1.758	60.258	0.01
	primary tumor	6.910	1.220	39.128	0.029

ReTURBT—repeat (restaging) transurethral resection of bladder tumor; HR—hazard ratio; CI—confidence interval.

## Data Availability

The data are available from the corresponding author upon reasonable request.

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
