# Peer review of "Assessing the Predictive Accuracy of EORTC, CUETO and EAU Risk Stratification Models for High-Grade Recurrence and Progression after Bacillus Calmette–Guérin Therapy in Non-Muscle-Invasive Bladder Cancer"

_cancers, 2024, doi:10.3390/cancers16091684_

Round 1
Reviewer 1 Report
Comments and Suggestions for Authors
The article addresses an important and current issue in the management of NMIBC, particularly the evaluation of the accuracy of existing risk stratifications (EORTC, CUETO, EAU) in predicting high-grade recurrence and progression post Bacillus therapy. The study is novel as it focuses on high-grade recurrences specifically, which have not been distinctly analyzed in previous models. This specific focus is crucial as high-grade recurrences have different prognostic implications compared to low-grade ones, highlighting a gap in current clinical practices and models.
The study effectively uses retrospective data from 171 NMIBC patients, providing a substantial basis for statistical analysis and model validation.
- Model Validation: Although the study introduces and validates new risk factors, the predictive accuracy of the models (C-index values) remains moderate. Enhancing these models with additional predictors or using more advanced statistical techniques might improve accuracy.
- Sample Size and Generalizability: The study is limited by its retrospective nature and the sample size from a single tertiary center, which may affect the generalizability of the findings. This should be highlighted in limitations and expanding the study to include multicentric data enhancing the robustness and applicability of the results in future studies should be declared.
- Detailed Risk Stratification: The study could benefit from a more detailed stratification of risk models, particularly by integrating patient demographics and treatment variables that could affect prognosis.
- Discussion: a deeper discussion focus must be provided. For instance, recent findings indicate that a specific bacterium on microbiome could be used as a specific biomarker of BCa risk, especially in older male patients (Reference to PMID: 38298766). Integrating microbiological data could potentially enhance the predictive power of current models and offer a broader understanding of the factors contributing to cancer progression and recurrence. This interesting result should be discussed and also urobiome profiling should be considered aspossible future integration into your risk models for NMIBC. In addition to the clinical and pathological factors currently used to predict recurrence and progression in NMIBC, future studies might consider evaluating systemic immune markers such as the systemic immune-inflammation index (SII). Recent research has demonstrated that elevated preoperative SII levels are associated with poorer oncological outcomes, including recurrence and overall survival, in patients undergoing radical cystectomy for bladder cancer (Reference to PMID: 38138166). Assessing SII could provide additional prognostic value and help refine risk stratification models for NMIBC, potentially identifying patients at higher risk for progression or recurrence. Please cite and include in your discussion.
The manuscript contributes valuable insights into the predictive modeling of NMIBC progression and recurrence, particularly for high-grade cases. It highlights significant limitations in current risk stratification models and proposes improvements that could aid in better patient management. However, further refinement and validation of the proposed models are essential to enhance their predictive accuracy and clinical utility. The study is a commendable step toward tailored therapeutic strategies but requires additional evidence for broader clinical implementation. A deeper discussion is needed.
Author Response
The article addresses an important and current issue in the management of NMIBC, particularly the evaluation of the accuracy of existing risk stratifications (EORTC, CUETO, EAU) in predicting high-grade recurrence and progression post Bacillus therapy. The study is novel as it focuses on high-grade recurrences specifically, which have not been distinctly analyzed in previous models. This specific focus is crucial as high-grade recurrences have different prognostic implications compared to low-grade ones, highlighting a gap in current clinical practices and models.
The study effectively uses retrospective data from 171 NMIBC patients, providing a substantial basis for statistical analysis and model validation.
Author Response: Thank you for your thoughtful and encouraging feedback. We sincerely appreciate your recognition of the importance of our study in addressing current challenges in NMIBC management, particularly in evaluating the accuracy of existing risk stratifications following BCG therapy. We attempted to introduce the required corrections and answer all of the posed questions. We hope that methodological details and additional statistics provided within our answers will be considered sufficient. Modified and added sentences were marked in red in the manuscript.
- Model Validation: Although the study introduces and validates new risk factors, the predictive accuracy of the models (C-index values) remains moderate. Enhancing these models with additional predictors or using more advanced statistical techniques might improve accuracy.
- Sample Size and Generalizability: The study is limited by its retrospective nature and the sample size from a single tertiary center, which may affect the generalizability of the findings. This should be highlighted in limitations and expanding the study to include multicentric data enhancing the robustness and applicability of the results in future studies should be declared.
Author Response: Thank you for your insightful comments and suggestions. We acknowledge the moderate predictive accuracy of the models and understand the potential for improvement through the inclusion of additional predictors or employing more advanced statistical techniques. Regarding sample size and generalizability, we agree that our study's retrospective design and single-center data limit the broader applicability of our findings. Following your suggestion we have duly noted these limitations and recognize the importance of expanding future studies to include multicentric data to enhance the robustness and generalizability of our results. We have included additional sentences in the limitations section to address these concerns.
Additional sentences were added to improve the discussion:
"Furthermore, future studies could incorporate emerging urine biomarkers and extended pathological assessment of TUR specimens including T1 sub-staging and immune-related gene expression to refine predictive models and enhance their clinical utility. Additionally, conducting multicenter validation studies will be imperative to confirm the generalizability and reliability of our findings across diverse clinical settings and within larger cohorts."
- Detailed Risk Stratification: The study could benefit from a more detailed stratification of risk models, particularly by integrating patient demographics and treatment variables that could affect prognosis.
- Discussion: a deeper discussion focus must be provided. For instance, recent findings indicate that a specific bacterium on microbiome could be used as a specific biomarker of BCa risk, especially in older male patients (Reference to PMID: 38298766). Integrating microbiological data could potentially enhance the predictive power of current models and offer a broader understanding of the factors contributing to cancer progression and recurrence. This interesting result should be discussed and also urobiome profiling should be considered aspossible future integration into your risk models for NMIBC. In addition to the clinical and pathological factors currently used to predict recurrence and progression in NMIBC, future studies might consider evaluating systemic immune markers such as the systemic immune-inflammation index (SII). Recent research has demonstrated that elevated preoperative SII levels are associated with poorer oncological outcomes, including recurrence and overall survival, in patients undergoing radical cystectomy for bladder cancer (Reference to PMID: 38138166). Assessing SII could provide additional prognostic value and help refine risk stratification models for NMIBC, potentially identifying patients at higher risk for progression or recurrence. Please cite and include in your discussion.
Author Response: Thank you for your thoughtful comments and suggestions to expand our discussion. We acknowledge the value of novel blood-based and easily available biomarkers such as SII and the most widely used NLR which were not validated in this study. We aimed to validate available risk tools such as EORTC and CUETO and none of them incorporates SII or another immune-inflammatory marker. Bladder and urine microbiome requires further elucidation and its role is largely unknown in the context of bladder cancer course. Recent studies sparked our interest in this area and its association with NMIBC outcomes. As we strongly agree with the importance of these topics we expanded the discussion following your suggestion. The lack of the above variables in the current models also reflects their limitations and the necessity to pursue new ones. We added a separate paragraph describing the role of the microbiome with respective and suggested citations and mentioning the role of blood-based biomarkers.
The following paragraph was added to the discussion:
“We anticipate the imminent update of current risk models as our understanding of the role of the urinary microbiome expands, and as urine and blood-based biomarkers are developed. Recent studies have underscored the potential significance of the urinary microbiome in the detection and course of NMIBC, sparking further interest in the investigation [23,24]. However, novel blood-based and easily accessible biomarkers, such as systemic immune-inflammatory markers and the well-recognized neutrophil-to-lymphocyte ratio, were not validated in this study due to their absence in the current clinically utilized risk models [21,25]. The suboptimal accuracy of existing models and the lack of inclusion of newly developed and potentially significant prognostic factors highlight the limitations of these models and emphasize the imperative for the development of new, more comprehensive risk assessment tools.”
The manuscript contributes valuable insights into the predictive modeling of NMIBC progression and recurrence, particularly for high-grade cases. It highlights significant limitations in current risk stratification models and proposes improvements that could aid in better patient management. However, further refinement and validation of the proposed models are essential to enhance their predictive accuracy and clinical utility. The study is a commendable step toward tailored therapeutic strategies but requires additional evidence for broader clinical implementation. A deeper discussion is needed.
Author Response: Thank you for your significant input and kind notice. We greatly appreciate your recognition of the valuable insights our manuscript offers into predictive modeling for NMIBC progression and recurrence. We believe that the introduced changes significantly improved the manuscript.
Reviewer 2 Report
Comments and Suggestions for Authors
Congratulations on the present paper!
There are few aspects that may improvethe paper’s quality and need further explanations.
1. Describe the techniques used for TURBT.
2. Were there any differences between en-block resection or classic technique.
3. Please add reference for the BCG administration scheme used in this study.
4. Please add citations for follow-up paragraph.
5. The reference list is too short for this type of study. Please find more relevant data to compare and extend the reference list.
6. There are also lots of paragraphs without citation which should be revised
Comments on the Quality of English Language
Minor Enlish issues
Author Response
Congratulations on the present paper!
There are few aspects that may improve the paper’s quality and need further explanations.
Author Response: We are sincerely grateful for the supportive comments and insightful suggestions provided by the Reviewer. We have diligently incorporated the necessary corrections and addressed all of the posed questions. We trust that the details and additional statistics provided within our responses adequately address any concerns. Modified and added sentences have been clearly marked in red within the manuscript.
- Describe the techniques used for TURBT.
Author Response: Thank you for this important suggestion. Additional sentences were added to the methods section.
“TURBT was performed with the patient in a lithotomy position under spinal or general anesthesia in accordance with EAU clinical guidelines [1]. A resection loop with monopolar current was utilized.”
- Were there any differences between en-block resection or classic technique.
Author Response: Thank you for this relevant inquiry. The majority of TURBTs were performed using classic techniques and resection of tumors with pieces. In 31 TURBT en-block technique was utilized at the discretion of the treating surgeon. As this was not assessed in any of studies that developed the scoring systems and risk models for NMIBC, we also decided not to elaborate on this in our paper which validates above mentioned risk models. Following your suggestion, we checked the effect of the en-block technique on survival outcomes. We did not find any differences between the conventional technique and en-block resection in terms of further RFS, HG RFS and PFS.
- Please add reference for the BCG administration scheme used in this study.
Author Response: In response to your valuable comment, we have ensured that proper citations have been added to the follow-up section of the methods. The BCG administration scheme used in this study is now referenced according to the EAU clinical guidelines for NMIBC which based on the results of RCTs. Additional sentences were added to methods section indicating the BCG regimen used.
Additional citations of RCTs that led to the establishment of the BCG schedule were added to the manuscript.
Lamm DL, Blumenstein BA, Crissman JD, Montie JE, Gottesman JE, Lowe BA, et al. Maintenance bacillus Calmette-Guerin immunotherapy for recurrent TA, T1 and carcinoma in situ transitional cell carcinoma of the bladder: a randomized Southwest Oncology Group Study. J Urol. 2000 Apr;163(4):1124–9.
Oddens J, Brausi M, Sylvester R, Bono A, van de Beek C, van Andel G, et al. Final results of an EORTC-GU cancers group randomized study of maintenance bacillus Calmette-Guérin in intermediate- and high-risk Ta, T1 papillary carcinoma of the urinary bladder: one-third dose versus full dose and 1 year versus 3 years of maintenance. Eur Urol. 2013 Mar;63(3):462–72
- Please add citations for follow-up paragraph.
Author Response: In response to your feedback, we have ensured that proper citations have been added to the follow-up section of the methods. The citation of EAU clinical guidelines for NMIBC was included to provide additional support for the methodology employed in our research. The follow-up schedule was derived from EAU recommendations. Additionally, both of the RCTs above cited and published by Lamm and Oddens support such a follow-up regimen.
- The reference list is too short for this type of study. Please find more relevant data to compare and extend the reference list.
Author Response: The reference list was updated with other relevant citations. Thank you for this comment.
- There are also lots of paragraphs without citation which should be revised
Author Response: Thank you for this relevant suggestion. We acknowledge the importance of ensuring proper citation throughout the manuscript. In response to your feedback, we have carefully reviewed the paragraphs lacking citations and have made necessary revisions to ensure that all statements are appropriately supported by references.
Thank you for your comments. We trust that these revisions address your concerns and enhance the credibility of our research.
Reviewer 3 Report
Comments and Suggestions for Authors
Dear Authors,
I read with interest your article, however I have some points to be better clarified:
1) In methods please clarify inclusion and exclusion criteria, the first paragraph suits better in the results section
2) SWOG regimen include 6 instillations of BCG at induction, not 5. How many patients had 5 instillations? Why? Do you think it might affect the results? Please expand in discussione
3) Was a power analysis conducted prior to the study? Or the timeframe and thus the sample was defined arbitrarily?
4) How many patients had a full BCG course? One year maintenance was far less effective than full schedule in terms of recurrence rate as reported by Oddens et al. Please expand this point in discussion
Author Response
Dear Authors,
I read with interest your article, however I have some points to be better clarified
Author Response: We are very grateful for all the supportive comments and suggestions provided by the Reviewer. We attempted to introduce the required corrections and answer all of the posed questions. We hope that details and additional statistics provided within our answers will be considered sufficient. Modified and added sentences were marked in red in the manuscript.
1) In methods please clarify inclusion and exclusion criteria, the first paragraph suits better in the results section
Author Response: We are thankful to the Reviewer for this notice. We appreciate your suggestion to clarify the inclusion and exclusion criteria in the methods section, and we made these adjustments accordingly. Please see the corrected versions of the methods section and additional sentences in the results section.
“Inclusion criteria included the presence of intermediate-, high-, and very-high-risk NMIBC in adult patients, treated with TURBT between 2010 and 2019 who received subsequent intravesical BCG instillations. Exclusion criteria encompassed lack of adequate induction course of BCG defined as at least 5 of 6 instillations (n=11), delayed BCG therapy > 4 months after the last TURBT (n=7), presence of isolated carcinoma in situ (CIS/Tis) without concomitant papillary tumor (n=27).”
2) SWOG regimen include 6 instillations of BCG at induction, not 5. How many patients had 5 instillations? Why? Do you think it might affect the results? Please expand in discussion
Author Response: Thank you for this important suggestion. We admit that eleven patients received 5 of 6 instillations of induction BCG due to adverse events. In 6 of these 11 patients, BCG was continued in further maintenance instillations as adverse events resolved. The remaining 5 patients abandoned BCG due to treatment intolerance. However, EAU guidelines and IBCG consensus approves the administration of 5 out of 6 instillations of induction courses and considers as adequate when at least one additional BCG course is given. Such scenarios reflect the daily clinical practice and our real-world data reflects the standard treatment patterns.
The discussion was expanded considering your suggestions in the query number 2 and 4:
“Another limitation that must be acknowledged is the suboptimal regimen duration, which nonetheless reflects real-world clinical practice. Eleven patients received 5 out of 6 instillations of induction BCG due to adverse events. In 6 of these 11 patients, BCG was continued in further maintenance instillations as adverse events resolved. Another important issue, considering the recommended regimen duration, is the low percentage (13.5%) of patients who received the 3-year maintenance schedule. Notably, even in RCTs like the SWOG study, which showed the superiority of maintenance BCG over induction alone, only 16% of patients completed the 3-year maintenance schedule. On the other hand, in the EORTC-GU Cancers Group Randomized Study, 35% of patients completed the 3-year full BCG maintenance regimen to which they were allocated. Our results reflect real-world treatment patterns, which provide valuable validation of available risk models, and such validation is necessary to ensure the applicability of the risk models beyond a clinical trial setting.”
3) Was a power analysis conducted prior to the study? Or the timeframe and thus the sample was defined arbitrarily?
Author Response: Thank you for your inquiry. While a formal power analysis was not conducted prior to the study, we would like to highlight that our research stems from a validation study encompassing all patients treated within a single tertiary center, renowned for its expertise in bladder cancer care. The timeframe for patient inclusion, spanning from 2010 to 2019, was determined based on the availability of comprehensive data and the center's clinical experience in managing such cases. Additionally, it's worth noting that the extensive follow-up period, extending to December 2023, provides valuable insights, especially considering the relatively lengthy duration of follow-up compared to other studies. With a sufficient number of events of interest for the validation calculations, our study included 73 (42.7%) recurrences, and 29 (17%) progressions to MIBC following BCG therapy.
4) How many patients had a full BCG course? One year maintenance was far less effective than full schedule in terms of recurrence rate as reported by Oddens et al. Please expand this point in discussion
Author Response: We are thankful for this thoughtful and kind suggestion. The respective paragraph was added to the discussion section. Notably, even in RCTs like the SWOG study, which showed the superiority of maintenance BCG over induction alone, only 16% of patients completed the 3-year maintenance schedule. We also included a low percentage (13.5%) of patients who received the 3-year maintenance schedule. Our results reflect real-world treatment patterns, which provide valuable validation of available risk models, and such validation is necessary to ensure the applicability of the risk models beyond a clinical trial setting. Please see the extended discussion paragraph in the response to your question number 2.
We are thankful for all the suggestive comments that were helpful in improving our manuscript.
Round 2
Reviewer 1 Report
Comments and Suggestions for Authors
the revised version is now worth of publication